# Antiviral Effect of Manganese against Foot-and-Mouth Disease Virus Both in PK15 Cells and Mice

**DOI:** 10.3390/v15020390

**Published:** 2023-01-30

**Authors:** Zhixiong Zhang, Rui Zhang, Juanbin Yin, Shuaiyang Zhao, Xiaodong Qin, Fei Chen, Yang Yang, Ling Bai, Zijing Guo, Yongshu Wu, Yanmin Li, Zhidong Zhang

**Affiliations:** 1State Key Laboratory of Veterinary Etiological Biology, Lanzhou Veterinary Research Institute, Chinese Academy of Agricultural Sciences, NO1 Xujiaping, Yanchangbu, Lanzhou 730046, China; 2College of Animal Husbandry and Veterinary Medicine, Southwest Minzu University, Chengdu 610041, China

**Keywords:** foot-and-mouth disease virus, Mn^2+^, antiviral effect, cytokines

## Abstract

Foot-and-mouth disease (FMD) is an acute contagious disease of cloven-hoofed animals such as cattle, pigs, and sheep. Current emergency FMD vaccines are of limited use for early protection because their protective effect starts 7 days after vaccination. Therefore, antiviral drugs or additives are used to rapidly stop the spread of the virus during FMD outbreaks. Manganese (Mn^2+^) was recently found to be an important substance necessary for the host to protect against DNA viruses. However, its antiviral effect against RNA viruses remains unknown. In this study, we found that Mn^2+^ has antiviral effects on the FMD virus (FMDV) both in PK15 cells and mice. The inhibitory effect of Mn^2+^ on FMDV involves NF-κB activation and up-regulation of interferon-stimulated genes. Animal experiments showed that Mn^2+^ can be highly effective in protecting C57BL/6N mice from being infected with FMDV. Overall, we suggest Mn^2+^ as an effective antiviral additive for controlling FMDV infection.

## 1. Introduction

Foot-and-mouth disease virus (FMDV) is the causative agent of FMD that can cause a significant epidemic disease threatening the livestock industry [1]. The FMDV has a broad host range of all cloven-hoofed animals including domestic and wild ruminants and pigs [2]. The FMDV is a member of the genus *Aphthovirus* of the family Picornaviridae, with a single-stranded positive-sense RNA genome. Seven serotypes of FMDV (O, A, C, SAT1, SAT2, SAT3, and Asia1) and multiple subtypes are currently recognized. The viral genome is approximately 8000 nucleotides in length, containing a single open reading frame that encodes four structural proteins (VP1, VP2, VP3, VP4) and eight nonstructural proteins (NSPs) (L^pro^, 2A, 2B, 2C, 3A, 3B, 3C^pro^, and 3D^pol^) [3]. The ability of the FMDV to spread rapidly in susceptible animals has made it a fairly serious disease classified by the World Organization for Animal Health (WOAH). Outbreaks of FMD can be devastating to the agricultural industry due to the expense of implementing control measures and the consequent trade restrictions that prevent the export of animals and animal products [4]. It is estimated that FMD costs between $6.5–21 billion per year in endemic regions [5]. Although inactivated FMD vaccines have been available since the early 1900s and new novel vaccines are being continuously developed, they offer little or no cross-protection against various serotypes of FMDV [6,7]. In addition, the current use of emergency FMD vaccines for inducing early protection is limited because the protection takes effect at least 7 days after vaccination [8,9]. Therefore, alternative approaches to antiviral agents are necessary to reduce the spread of FMDV during the epidemic.

Manganese (Mn) is one of the most abundant metals in mammalian tissues, ranging from 0.3 to 2.9 mg/kg wet tissue weight, and is required for various physiological processes including development, reproduction, neuronal function, immune regulation and antioxidant defense [10,11]. Manganese acts by regulating various manganese-dependent enzymes, including oxidoreductases, isomerases, transferases, ligases, lyases, and hydrolases. It is also a basic component of some metalloenzymes such as manganese superoxide dismutase (SOD2), glutamine synthetase (GS) and arginase [12]. The Mn^2+^ ion is essential for host protection against DNA virus infection by improving the sensitivity of the DNA sensor cGAS and its downstream adaptor protein STING [13]. However, there are few studies on the role of Mn^2+^ in RNA virus infection, and its mechanism of action is still unclear.

In this study, we found that Mn^2+^ had antiviral effects against FMDV both in PK15 cells and mice. The inhibitory effect of Mn^2+^ on FMDV involves NF-κB activation and up-regulation of interferon-stimulated genes. Animal experiments further demonstrate that Mn^2+^ is highly effective in protecting C57BL/6N mice infected with FMDV. Altogether, these results provide a potential application for a treatment strategy for controlling FMDV infection.

## 2. Materials and Methods

### 2.1. Reagents and Antibodies

The MnCl_2_(H_2_O)_4_ was purchased from Sigma Aldrich (CAS: 13446-34-9). The rabbit polyclonal antibodies anti-phospho-NF-κB (93H1; catalog number 3033S), and anti-phospho-TBK1/NAK XP (Ser172 and D52C2; catalog number 5483S), and anti-TBK1/NAK (E8I3G; catalog number 38066s), and mouse monoclonal antibody anti-NF-κB (L8F6; catalog number 6956T) were from Cell Signaling Technology. Mouse monoclonal antibody anti-β-tubulin (catalog number 66240-1-Ig) was obtained from Proteintech. Rabbit Polyclonal antibody anti-FMDV VP1 protein (type O) (bs-41049R) was purchased from Bioss. The HRP Goat Anti-Rabbit IgG secondary antibody (catalog number AS014) and goat anti-mouse IgG secondary antibody (catalog number AS003) were from ABclonal.

### 2.2. Cell Culture and Maintenance

The BHK-21 (baby hamster kidney, ATCC CCL-10) and PK15 (porcine kidney, ATCC CCL-33) cells were all grown in Dulbecco’s modified Eagle’s medium (DMEM; Gibco, C14190500BT) supplemented with 10% FBS (BI, 04-001-1ACS), 100 U/mL penicillin and 50 μg/mL streptomycin (Gibco, 15140–122). Cells were grown in monolayers in tissue culture flasks or dishes at 37 °C under 5% CO_2_.

### 2.3. Virus

The FMDV O strain was obtained from OIE/National FMD Laboratory (Lanzhou, China). To prepare viral stock, adsorption of FMDV on BHK-21 cells at 37 °C for 1 h, unbound viruses were removed by washing three times with PBS, then cells were coated with DMEM containing 2% FBS and incubated at 37 °C for 8 h. Next, they were frozen and thawed three times repeatedly, before the viral stock was clarified by centrifugation at 8000× *g* for 10 min. The titer was determined by the Reed-Muench method and expressed as 50% tissue culture infectious does (TCID_50_).

### 2.4. Mn^2+^ Cytotoxicity Assay in PK15 Cells

The PK15 cells were seeded into 96-well plates at a density of 3 × 10^5^/mL for culture, with 200 µL per well. The following day, the cell culture supernatant was removed, and 100 µL of 2-fold serial dilution gradient dilution of the Mn^2+^ was added to each well, the final concentrations of the Mn^2+^ were 3.9, 7.81, 15.63, 31.25, 62.5, 125, 250, 500 and 1000 µM, respectively. Three replicate wells were used for each concentration. At the same time, a blank control group was set up. Cells were incubated at 37 °C under 5% CO_2_. After 72 h, 100 µL of CellTiter-Glo cell activity assay reagent (Promega) was added to each well. The fluorescence signal of the cells was detected under 560 nm excitation light. Cytotoxicity is expressed as 50% cytotoxic concentration (CC_50_), which is defined as the concentration required to reduce cell viability by 50% of the control value. GraphPad 8 software was used for regression analysis of the data for determining CC_50_ of Mn^2+^ on PK15 cells.

### 2.5. Western Blotting

After treatment, whole cells or organelle fractions were harvested and lysed at 4 °C in an ice-cold Cell Lysis Buffer (Beyotime) containing a protease inhibitor cocktail (Beyotime). A bicinchoninic acid (BCA) assay (Thermo Fisher Scientific) was used to detect protein concentration. Protein samples were subjected to SDS-PAGE and transferred onto a polyvinylidene difluoride (PVDF) membrane (GE). The membranes were blocked with 5% nonfat milk for 1 h at room temperature. Subsequently, the membrane and specific primary antibody were incubated overnight at 4 °C, followed by incubation with horseradish peroxidase (HRP)-labeled secondary antibodies for 1 h at room temperature. The membrane was then subjected to an ECL assay.

### 2.6. RNA Extraction and RT-qPCR

Total RNA was extracted from each sample with Trizol reagent (Invitrogen) and quantified before the RNA was used for reverse transcription (RT) with a PrimeScript RT reagent kit (TaKaRa) following the manufacturer’s protocol. Real-time PCR was performed with 1 µL cDNA in 10 µL with SYBR Green master mix (QIAGEN) according to the manufacturer’s instructions. The data were presented as an accumulation index (2^−△△Ct^). Sequences of the primers used for PCR are listed in a Appendix A (Appendix A). Relative gene expression levels were normalized to a GAPDH control.

### 2.7. Animal Infection and Preparation of Samples

C57BL/6N mice at age of five weeks were obtained from the Animal Center for Lanzhou Veterinary Research Institute (LVRI). Animals had free access to water and food. The animal experiments were performed at the biosafety level 3 laboratory of Lanzhou Veterinary Research Institute, Chinese Academy of Agricultural Sciences. C57BL/6N mice were inoculated with 1 mg/kg or 2 mg/kg Mn^2+^ by intravenous (I.V.) injection at 12 h before FMDV infection. The PBS was inoculated by I.V. injection at 12 h before FMDV infection, as the negative control. Mice were challenged with 2 × 10^5^ TCID_50_ of FMDV by intraperitoneal (I.P.) injection. The mice were monitored daily for 5 days. The animals were euthanized when infected mice showed severe respiratory distress, inability to maintain body temperature, or secondary infection. At 48 h post-inoculation (hpi), mice were euthanized, and tissues of the heart, spleen, and pancreas were collected and stored at −80 °C until use.

### 2.8. Detection of Cytokine Levels in the Serum

C57BL/6N mice were inoculated with 2 mg/kg Mn^2+^ intravenous (I.V.) injection at 12 h before FMDV infection. Mice were challenged with 2 × 10^5^ TCID_50_ of FMDV by I.P. injection. The serum was isolated from the whole blood obtained from the infraorbital sinus of the mice at 24 and 48 hpi. The IFN-α, IFN-β, IL-6, CCL2 and CCL5 enzyme linked immunosorbent assay (ELISA) kits were purchased from R&D Systems, and the level of cytokines in the serum was measured according to the manufacturer’s instructions. The concentration of each protein was determined by interpolation of the standard curves.

### 2.9. Statistical Analysis

The data presented in this paper are expressed as means and standard deviations (SD) for at least two replicates and were evaluated with a two-tailed student’s *t*-tests or two-way analysis of variance (ANOVA) in GraphPad Prism software (version 8.0).

## 3. Results

### 3.1. Antiviral Activity of Mn^2+^ against FMDV Infection in PK15 Cells

To assess the toxicity of Mn^2+^ on cells susceptible to FMDV, PK15 cells were treated with 2-fold serial dilution of the Mn^2+^. After 72 h of incubation, cell viability was measured and compared with untreated cell controls. As shown in Figure 1A, the cytotoxic effect of Mn^2+^ on PK15 cells was dose dependent and the cytotoxic concentration (CC_50_) of Mn^2+^ on PK15 cells was calculated to be 438 µM.

To investigate the antiviral effect of Mn^2+^ on FMDV infection, PK15 cells were treated with different concentrations of Mn^2+^ (0, 10, 20, 50, 100 and 200 µM) and infected with FMDV at 1.0 MOI. At 10 hpi, the viral titers (Figure 1B) and viral RNA level (Figure 1C) were then analyzed by TCID_50_, and real-time RT-PCR. As shown in Figure 1, compared with the untreated FMDV control group, the viral titers and viral RNA level significantly decreased in the cells treated with different concentrations of Mn^2+^ (ranging from 20 to 200 μM) (*p* < 0.01), indicating that Mn^2+^ had antiviral activity on FMDV in a dose-dependent manner.

To further explore the effect of Mn^2+^ on the replication of FMDV, a time-of-addition test was performed. The PK15 cells were treated with 50 μM Mn^2+^ at 1 h before FMDV infection (I), at the time of FMDV infection (II), 2 h (III), and 4 h (IV) after FMDV infection (Figure 1D). As shown in Figure 1E, regardless of different treatment time point with Mn^2+^, the FMDV titers and viral RNA level were significantly lower than that in the untreated group (*p* < 0.001). Moreover, the antiviral effect of Mn^2+^ before virus infection (I) and in the virus entry phase (II) was better than that in 2 h (III) and 4 h (IV) post-treatment groups.

### 3.2. Mn^2+^ Activates Anti-Viral Innate Immunity via the NF-κB Signaling Pathway in PK15 Cells

To evaluate the mechanism for the antiviral effect of Mn^2+^ in PK15 cells, PK15 cells were treated with Mn^2+^ for 1 h in advance and then infected with FMDV. Western Blot analysis showed that the levels of P-P65 and P-TBK1 in Mn^2+^ treated PK15 cells were significantly up-regulated compared with those in the control group, while the expression of FMDV VP1 was significantly decreased (Figure 2A,B). In addition, measurement of Mn^2+^-induced ISGs and interferon (IFN) expression in PK15 cells by RT-qPCR showed that the mRNA levels of genes encoding IFN-α, IFN-β, myxovirus resistance (Mx), 2′-5′ oligoadenylate synthetase (OAS), ISG54, and CCL2 in Mn^2+^ treated cells were significantly higher than those in the control group (Figure 3A–F).

### 3.3. Enhanced Survival Rate of Mice Inoculated with Mn^2+^ after Being Challenged by FMDV

To investigate the in vivo antiviral activity of Mn^2+^, 5 weeks of mice (*n* = 10) were intravenously injected with 1 mg/kg or 2 mg/kg Mn^2+^ 12 h before mice were challenged with the virus with a lethal dose of of 2 × 10^5^ TCID_50_ and mice (*n* = 10) inoculated with PBS as the untreated-control. The mice were daily monitored for 5 days. These virus-infected mice had significantly longer survival times after Mn^2+^ pretreatment compared to the PBS control group. The results showed that 60% (6/10) and 80% (8/10) of mice pre-treated with Mn^2+^ at 1 and 2 mg/kg respectively still survived at 84 hpi, but all PBS treated mice died at this time point (Figure 4A). A further analysis showed that viral RNA level significantly reduced in the heart (Figure 4B), spleen (Figure 4C), and pancreas (Figure 4D) collected at 48 hpi from the mice pretreated with Mn^2+^ at 2 mg/kg. This demonstrated that Mn^2+^ could enhance mice to resist FMDV infection.

### 3.4. Cytokine Production in Mice Treated with Mn^2+^

Previous studies have reported upregulated relative expressions of some cytokines in Mn^2+^ treated cells [13]. To investigate whether Mn^2+^ functions similarly in vivo, the cytokine levels were measured in mice inoculated with Mn^2+^ at 2 mg/kg by intravenous (I.V.) injection at 12 h before FMDV infection and serum samples were collected at 24 and 48 hpi. The results showed that the IFN-α (Figure 5A), IFN-β (Figure 5B), IL-6 (Figure 5C), CCL2 (Figure 5D), and CCL5 (Figure 5E) protein levels were significantly increased at 24 and 48 hpi in comparison with PPS treated groups. Collectively, these results suggested that Mn^2+^ positively regulates the NF-κB signaling pathway and antiviral gene expression induced by FMDV infection in mice.

## 4. Discussion

Transition metals such as iron (Fe), manganese (Mn), copper (Cu), and zinc (Zn) are essential for all forms of life because 30% of enzymes require metal cofactors [14]. Clinical deficiency of Fe or Zn in the host increases infectious disease and mortality [15,16]. Manganese is one of the most abundant metals in the body, with normal concentrations ranging from 0.072 μM to 0.27 μM in the human blood [17] and from 20 μM to 53 μM in the human brain [18]. A study by Wang et al. suggested that Mn^2+^ is necessary for the host to resist DNA viruses by increasing the sensitivity of the DNA sensor cGAS and its downstream adaptor protein STING. The Mn^2+^ ion is released from membrane-enclosed organelles upon viral infection and accumulates in the cytoplasm where it binds directly to cGAS. The Mn^2+^ enhances the sensitivity of cGAS to double-stranded DNA (dsDNA) and its enzymatic activity so that cGAS can produce secondary messenger cGAMP in the presence of low concentrations of dsDNA, otherwise, it will be non-irritating. The Mn^2+^ ion also enhances STING activity by enhancing cGAMP-STING binding affinity. Mn-deficient mice showed reduced cytokine production and were more susceptible to DNA viruses, and STING-deficient mice did not show increased susceptibility. These findings suggest that Mn^2+^ is an important substance for host resistance against DNA viruses [13]. However, the role of Mn^2+^ in RNA virus infection is poorly investigated.

Foot-and-Mouth-Disease is a viral infection of livestock of critical socioeconomic importance. Field studies from areas of endemic FMD suggest that animals can be simultaneously infected by more than one distinct variant of FMDV, potentially resulting in emergence of novel viral strains through recombination [19]. Vaccine-mediated protection against clinical FMD correlates with induction of a quantifiable neutralizing antibody response, and vaccination campaigns are dependent upon close matching of vaccine strains with regionally circulating field strains for effective protection [20]. The FMD vaccines do not protect against subclinical infection of the upper respiratory tract of ruminants, where FMDV may cause early neoteric subclinical infection and may persist for months or years after virus exposure [21]. Consequently, it is important to take effective measures to control the virus. Researchers have proposed many approaches to control viral infections, including a range of broad-spectrum antivirals against host factors or direct antiviral agents capable of inhibiting multiple viruses [22]. In this study, we preliminary assessed the activity of Mn^2+^ against FMDV infection both in PK15 cells and mice; however, it needs to extend to evaluate manganese effect in swine as well as bovine.

Our findings showed that Mn^2+^ significantly inhibited FMDV replication in PK15 cells, particularly at concentrations ranging from 20 μM to 200 μM. Notably, Mn^2+^ induced increased cytotoxicity in PK15 cells at concentrations greater than 200 μM. Therefore, in order to minimize the toxic risk of Mn^2+^, careful monitoring and adjustment of the dose of Mn^2+^ in animals are required in the future. Importantly, when PK15 cells were exposed to Mn^2+^ at several points during the viral replication phase, we found that Mn^2+^ reduced viral replication both 1 h before and 4 h after viral infection, but that pretreatment with viral infection inhibited viral replication more significantly, indicating that the early timing of viral replication is the main target of Mn^2+^ antiviral effects. In addition, we found that Mn^2+^-treated PK15 cells were able to activate the NF-κB signaling pathway and upregulated ISGs gene expression. Collectively, these results support that Mn^2+^ has antiviral activity against FMDV.

Although Mn has been shown to inhibit FMDV replication at the cellular level, does it have an antiviral effect in vivo? In this study, five-week-old C57BL/6N mice were used to observe their biological activity in vivo. We found that mice treated with 1 mg/kg and 2 mg/kg Mn^2+^, respectively, significantly prolonged their survival time of mice compared with PBS mice, and about half of the mice did not die. In addition, the viral load in various tissues of mice decreased significantly after Mn^2+^ treatment, and the interferon level in the serum of mice increased. In the future, live experiments in cattle and pigs will be required to determine the real potential of Mn^2+^ as an adjunct control measure for FMD outbreaks.

In summary, we have shown that Mn^2+^ has antiviral effects against the FMDV (serotype O) in PK15 cells and mice. In addition, inhibition of FMDV replication occurs primarily early in infection. Further studies are necessary to better understand the exact mechanism of the action of Mn^2+^ against RNA virus infection. The antiviral effect of Mn^2+^ may provide a basis for further development of effective antiviral agents combined with other antiviral molecules or vaccines against FMD.

## Figures and Tables

**Figure 1 viruses-15-00390-f001:**
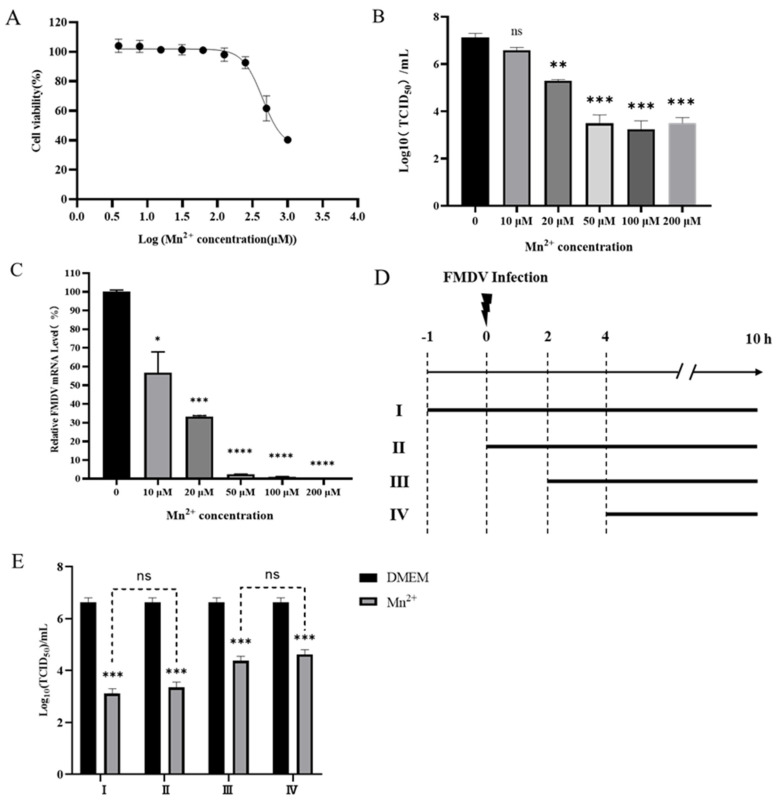
Antiviral activity of Mn^2+^ against FMDV in PK15 cells. (**A**) PK15 cells were incubated with 2-fold serial dilutions of Mn^2+^. After 72 h of culture, cell viability was measured and compared with the untreated cell control group. (**B**) PK15 cells were treated with 0, 10, 20, 50, 100, and 200 μM Mn^2+^, respectively, and infected with FMDV at MOI 1. The cells and supernatants were collected 12 h after FMDV infection, the viral titers were detected by TCID_50_, and the virus copy number was detected by real-time RT-PCR (**C**). (**D**) Schematic diagram of the time course of the Mn^2+^ addition experiment. (**E**) PK15 cells were treated with 50 μM Mn^2+^ at 1 h before FMDV infection (I), at the time of FMDV infection (II), 2 h (III), and 4 h (IV) after FMDV infection. The cells were infected with FMDV at MOI of 1 for 1 h. In each step, the cells were washed twice with PBS. The supernatants were collected at 10 h post-FMDV infection and virus titration was performed. Error bars indicate the standard deviation (SD) from the mean. A *t*-test was performed to identify statistically significant differences. *, *p* < 0.05; **, *p* < 0.01; ***, *p* < 0.001; ****, *p* < 0.0001.

**Figure 2 viruses-15-00390-f002:**
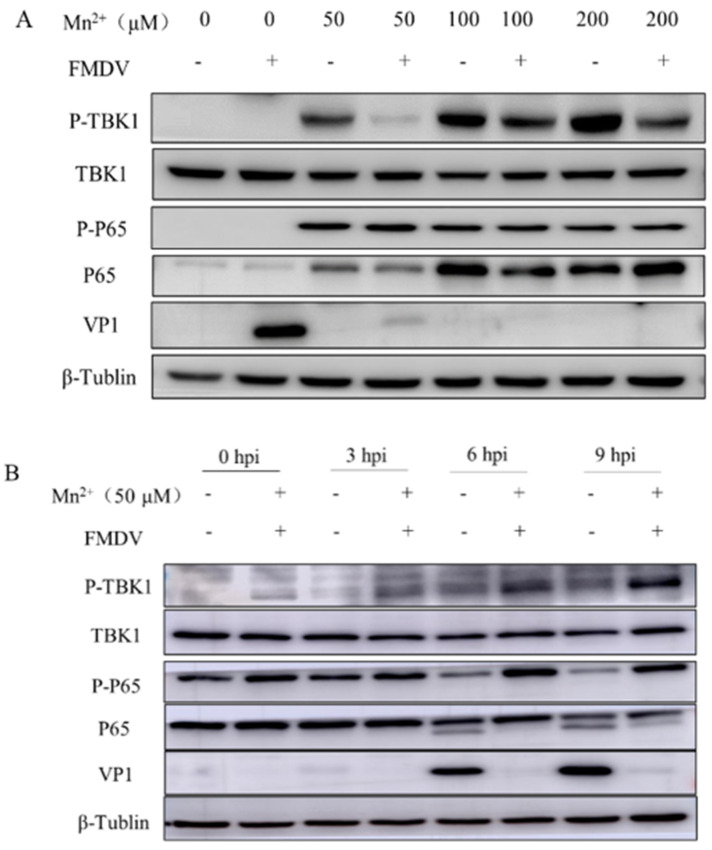
Activation of the NF-κB signaling pathway in PK15 cells treated with Mn^2+^. (**A**) PK15 cells were treated with 0, 50, 100, and 200 μM Mn^2+^, respectively, and PK15 cells were collected 8 h after FMDV infection for western blot assay. (**B**) PK15 cells were treated with 50 μM Mn^2+^, and PK15 cells were collected at 3, 6, and 9 h after FMDV infection for western blot assay.

**Figure 3 viruses-15-00390-f003:**
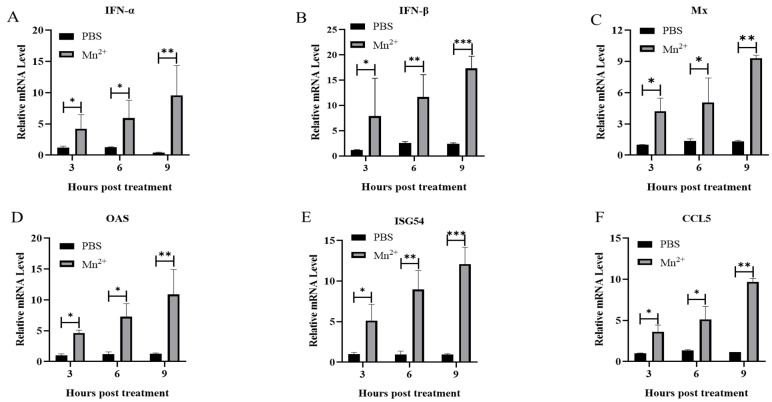
Activation of ISGs and IFN-I in PK15 cells treated with Mn^2+^. PK15 cells were treated with 50 μM Mn^2+^ and were collected at 3, 6, and 9 h following inoculation, and the mRNA levels of IFN-α (**A**), IFN-β (**B**), Mx (**C**), OAS (**D**), ISG54 (**E**) and CCL5 (**F**) were measured using RT-qPCR. A *t*-test was performed to identify statistically significant differences. *, *p* < 0.05; **, *p* < 0.01; ***, *p* < 0.001.

**Figure 4 viruses-15-00390-f004:**
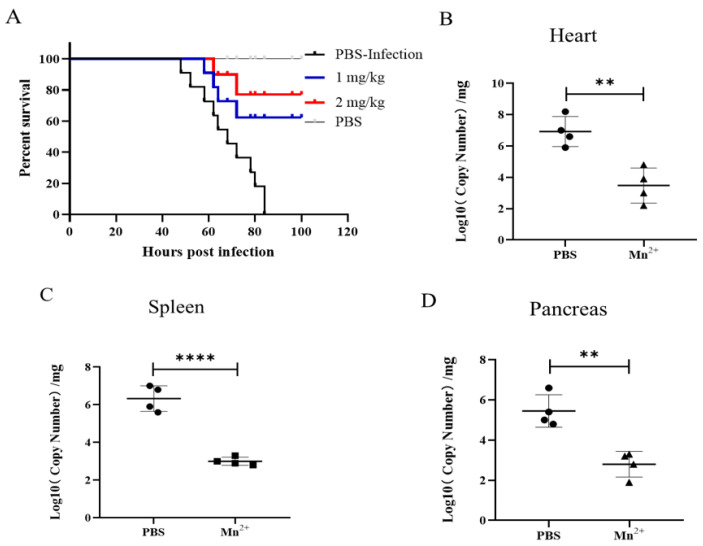
Survival rates of mice injected with Mn^2+^ after being challenged with FMDV. C57BL/6N mice were inoculated with 1 mg/kg or 2 mg/kg Mn^2+^ by intravenously (I.V.) injection at 12 h before FMDV infection. PBS was inoculated by I.V. injection at 12 h before FMDV infection, as the negative control. Mice were challenged with 2 × 10^5^ TCID_50_ of FMDV by I.P. injection. The mice were monitored for 5 days. (**A**) Mn^2+^ treated mice were partially resistant to FMDV-induced death (*n* = 10 per group). FMDV RNA was reduced in the heart (**B**), spleen (**C**), and pancreas (**D**) of FMDV-infected mice for 48 h following Mn^2+^ treatment (*n* = 4 per group). A *t*-test was performed to identify statistically significant differences. **, *p* < 0.01; ****, *p* < 0.0001.

**Figure 5 viruses-15-00390-f005:**
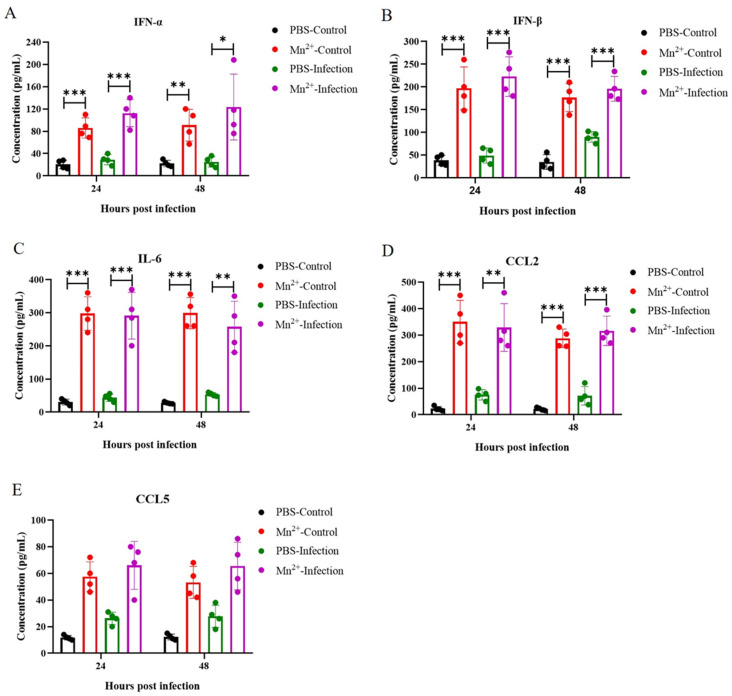
Induction of cytokines by Mn^2+^ in mice. C57BL/6N mice were inoculated with 2 mg/kg Mn^2+^ intravenously (I.V.) injection at 12 h before FMDV infection. Mice were challenged with 2 × 10^5^ TCID_50_ of FMDV by I.P. injection. Serum samples were collected at 24 and 48 h post-treatment and the protein levels of mouse IFN-α (**A**), IFN-β (**B**), IL-6 (**C**), CCL2 (**D**), and CCL5 (**E**) were measured using ELISA (*n* = 4 per group). A *t*-test was performed to identify statistically significant differences. * *p* < 0.05; ** *p* < 0.01, *** *p* < 0.001.

## Data Availability

Not applicable.

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
