# Peer review of "Antiviral Effect of Manganese against Foot-and-Mouth Disease Virus Both in PK15 Cells and Mice"

_viruses, 2023, doi:10.3390/v15020390_

Round 1
Reviewer 1 Report
In this study the authors showed the antiviral activity of Mn+2 on FMDV both on in vitro and in vivo. The results showed that Mn+2 reduced the viral load, increased the host antiviral immune responses and activated the NF-κB signaling pathway
Major comments
1) More details are required in vivo tests, how many animals used/ group. inoculum dose, infection period, ....etc.
2) The primers listed in the study were mainly for PK15 cells which were swine derived. Did the same primers work for mice???????
3) I would suggest confirm the NF-kb signaling pathways using mice tissues.
4) Rationale of using proinflammatory cytokines should be mentioned for examples, IL-1B. Cxcl-1, IFN-g,.... are not done, why?
5) I would prefer in figure 3 and figure 5 to display blot dots to verify how many experiments done/ or how many animals from which the data were extracted
Author Response
Dear Editor/Reviewers:
Thank you for kindly reviewing the manuscript entitled “Antiviral effect of Manganese against foot-and-mouth disease virus both in vitro and in vivo” (viruses-2112479). We have revised the manuscript as required and attached the revised versions with track changes. Thank you very much for your attention to our manuscript and guidance to improve our manuscript.
Reviewer 1:
Major comments:
1) More details are required in vivo tests, how many animals used/group. inoculum dose, infection period, etc.
Response: Thank you for your suggestion. A number of animals: n=10 per group, n=4 per group; Inoculum dose: 2 x 105 TCID50; …etc. It has been revised in the revised version of the manuscript (lines 135-138, lines 189-193, lines 247-249, line 253, line 255).
2) The primers listed in the study were mainly for PK15 cells which were swine derived. Did the same primers work for mice?
Response: The primers listed in this paper were designed with reference to swine-derived gene and was used only for the detection of cytokine mRNA in swine-derived PK15 cells.
3) I would suggest confirm the NF-kb signaling pathways using mice tissues.
Response: Due to the complexity of cell components in mouse tissues, the key molecules of the NF-kb signaling pathway are not well detected by western blotting.
4) Rationale of using proinflammatory cytokines should be mentioned for examples, IL-1β. Cxcl-1, IFN-g are not done, why?
Response: Mn-induced production of IFN-α, IFN-β, IL-6, CCL2, and CCL5 has been reported (Wang et al., 2018), so we performed detection of these cytokines. The information has been added in the revised version of the manuscript (lines 204-205).
5) I would prefer in figure 3 and figure 5 to display blot dots to verify how many experiments done/ or how many animals from which the data were extracted.
Response: Thank you for your suggestion. In Figure 3, we use the average value of RT-qPCR data, so a histogram is used for display. In Figure 5, we have modified the revised version following your comments.
Editors and reviewers are instrumental in improving the quality of articles. We have provided explanations and responses to all the reviewers' questions and comments. We would love to thank you for allowing us to resubmit a revised copy of the manuscript and we highly appreciate your time and consideration. We sincerely hope this manuscript will be finally acceptable to be published on Viruses. Thank you very much for all your help and looking forward to hearing from you soon.
Best regards
Sincerely,
Zhang Zhidong,
January 10, 2023
College of Animal Husbandry and Veterinary Medicine
Southwest Minzu University
Tel: +86-028-85528276
Email: zhangzhidong@swun.edu.cn
Address: No16 Yihuan Road, Chengdu, Sichuan, China

Reviewer 2 Report
The study evaluated the effec of Manganese on FMDV serotype O infection on PK and BHK cells as well as on mice model. The outhors claimed that manganese decreases FMDV replication in cells as well as on mice. The mechanism of protection was due the NFKb activation and up regulation of IFN related genes. The results are interesting howver the authors over reacted to the results and application of manganese. It should be clear on the article that FMDV is important to bovine trade and there are several strain of FMDV. Also it should be clear that this study was done in mice model and it's potential extension to bovine and swine should be further investigated.
Major:
1. Title, it should be change to: Antiviral effect of Manganese against foot-and-mouth disease virus in PK cells and mice. Otherwise, the information lead the readers to expect the effect of manganese in cows and swine.
2. Introduction line 32: Please include information regard the importance of FMDV in bovine.
3. Line 49: again, the conclusion should be more realistic: Manganese had antiviral effects against FMDV in PK cells and mice model.
4. The authors overreacted in the conclusion. It’s premature the affirmation: “All together, these results provide a novel treatment strategy for controlling FMDV infection.” Instead, it should be written “These results provide a potential application for treatment strategy for controlling FMDV infection”
5. The study was only done in FMDV O strain, how about other FMDV strain??
6. Line235: The authors need to add the information that FMDV is important to bovine production.
7. Line 240: Authors should be precautious with information and clarify that the evaluation of antiviral effect of Manganese was performed in mice only. This is only a preliminary evaluation; therefore, it needs to extend to evaluate manganese effect in swine as well as bovine.
8. Line 241: This affirmation is confused. Please correct to “Our findings showed that Mn2+ significantly inhibited FMDV replication in PK15 cells when mice were infected orally”.
9. Line 261: This information must be corrected to “In summary, we have shown that Mn2+ has antiviral effects against FMDV (serotype O) in vitro and in vivo (mice model).”
Author Response
Dear Editor/Reviewers:
Thank you for kindly reviewing the manuscript entitled “Antiviral effect of Manganese against foot-and-mouth disease virus both in vitro and in vivo” (viruses-2112479). We have revised the manuscript as required and attached the revised versions with track changes. Thank you very much for your attention to our manuscript and guidance to improve our manuscript.
Reviewer 2:
Major comments:
1.Title:It should be change to: Antiviral effect of Manganese against foot-and-mouth disease virus in PK15 cells and mice. Otherwise, the information lead the readers to expect the effect of manganese in cows and swine.
Response: Thank you for your suggestion. The title has been revised as suggested in the revised version of the manuscript.
2.Introduction line 32: Please include information regard the importance of FMDV in bovine.
Response: The information has been added in the revised version of the manuscript (lines 35 to 43).
3.Line 49: again, the conclusion should be more realistic: Manganese had antiviral effects against FMDV in PK cells and mice model.
Response: Thank you for your suggestion. It has been revised in the revised version of the manuscript (lines 56-57; line 312).
4.The authors overreacted in the conclusion. It’s premature the affirmation: “All together, these results provide a novel treatment strategy for controlling FMDV infection.” Instead, it should be written “These results provide a potential application for treatment strategy for controlling FMDV infection”.
Response: Thank you for your suggestion. It has been revised in lines 62-64 of the revised version of the manuscript.
5.The study was only done in FMDV O strain, how about other FMDV strain?
Response: The FMDV includes seven serotypes (O, A, C, SAT1, SAT2, SAT3, and Asia1). At present, the FMDV O is the predominant FMDV serotype causing outbreaks worldwide. Therefore, the FMDV O strain was used in this study. Our study indicated that Mn2+ could inhibit FMDV O infection in the cells and mice via induction of IFNs which have an inhibitory effect on FMDV A replication (CHINSANGARAM et al, 2001 and Zhang et al, 2002). We expected that such an antiviral effect of Mn2+ could occur to in other FMDV stains. But it needs to extend to study the antiviral effect of Mn2+ on FMDV strain in the future.
6.Line235: The authors need to add the information that FMDV is important to bovine production.
Response: The information that FMDV is important to bovine production has been included in lines 275-284 of the revised version of the manuscript.
7.Line 240: Authors should be precautious with information and clarify that the evaluation of antiviral effect of Manganese was performed in mice only. This is only a preliminary evaluation; therefore, it needs to extend to evaluate manganese effect in swine as well as bovine.
Response: Thank you for your suggestion. It has been revised in the revised version of the manuscript.
8.Line 241: This affirmation is confused. Please correct to “Our findings showed that Mn2+ significantly inhibited FMDV replication in PK15 cells when mice were infected orally”.
Response: We apologize for this inappropriate description. It has been revised in the revised version of the manuscript.
9.Line 261: This information must be corrected to “In summary, we have shown that Mn2+ has antiviral effects against FMDV (serotype O) in vitro and in vivo (mice model).”
Response: It has been revised in the revised version of the manuscript.
Editors and reviewers are instrumental in improving the quality of articles. We have provided explanations and responses to all the reviewers' questions and comments. We would love to thank you for allowing us to resubmit a revised copy of the manuscript and we highly appreciate your time and consideration. We sincerely hope this manuscript will be finally acceptable to be published on Viruses. Thank you very much for all your help and looking forward to hearing from you soon.
Best regards
Sincerely,
Zhang Zhidong,
January 10, 2023
College of Animal Husbandry and Veterinary Medicine
Southwest Minzu University
Tel: +86-028-85528276
Email: zhangzhidong@swun.edu.cn
Address: No16 Yihuan Road, Chengdu, Sichuan, China

Round 2
Reviewer 1 Report
The authors have replied my comments. No further questions